# Barriers to osteopathic manipulative medicine use: A qualitative study of physician attitudes and experiences

Stephen K. Stacey[1]*, Kelsie M. Prins[1], Sydney A. Anderson[1], Alessandro Quartiroli[2]

1 Department of Family Medicine, Mayo Clinic Health System – Southwest Wisconsin region, La Crosse, Wisconsin, United States of America, 2 Department of Psychology, University of Wisconsin–La Crosse, La Crosse, Wisconsin, United States of America

⊙ These authors contributed equally to this work.
* stacey.stephen@mayo.edu

## Abstract

### Purpose

Despite growing recognition of osteopathic manipulative medicine (OMM) as an effective treatment for chronic pain, little is known about the real-world barriers clinicians face when attempting to integrate OMM into their clinical practice. Fewer than half of practicing Doctors of Osteopathic Medicine (DOs) incorporate OMM into patient care. Understanding the barriers that hinder OMM use is critical to bridging this gap and expanding access to this evidence-based treatment.

### Methods

A survey was distributed to DOs from Mayo Clinic and Mayo Clinic Health System, inviting volunteers for semi-structured interviews about their attitudes and experiences with OMM. The interviews were audio-recorded, professionally transcribed, and analyzed using inductive reflexive thematic analysis. A thematic map was developed through investigator triangulation to connect key themes and subthemes.

### Results

Among 27 survey respondents who volunteered, 16 completed in-depth interviews, representing diverse clinical roles and experiences. Analysis identified six overarching themes: system, capability, utility, training, attitudes and beliefs, and resources. These themes were supported by direct quotes and linked through subthemes that illustrated nuanced barriers and facilitators influencing OMM integration.

### Conclusions

This study provides important qualitative insights into how DOs perceive the systemic, practical, and perceptual barriers that hinder implementation of OMM into

**Data availability statement:** Participants did not give consent for broad distribution of their interviews. Researchers wishing to obtain access to the interview transcripts may contact the Mayo Clinic Health System regional research team at mchsswwires@mayo.edu.

**Funding:** This project was supported by funding from a Mayo Clinic Health System Research Internal Award, awarded to SKS (no grant number available). The funding source had no involvement in study design; collection, analysis, and interpretation of data; writing of the report; or decision to submit the article for publication.

**Competing interests:** The authors have declared that no competing interests exist.

clinical practice. Time constraints, scheduling complexity, and limited patient awareness emerged as significant barriers, compounded by broader systemic issues such as patient panel management and practice autonomy. Addressing these barriers through intentional support from institutional leadership could improve access to this valuable treatment.

## Background

Osteopathic manipulative medicine (OMM) is an evidence-based treatment for several chronic pain conditions and is useful in situations where medications may be contraindicated or unwise [1–4]. However, it appears to be underutilized in current clinical practice [5,6]. Despite extensive OMM training during medical school, only 43% of DOs report offering OMM to their patients. Of those who offer it, about half provide it to fewer than 5% of their patients [7]. This infrequent use of OMM limits patient access to valuable physician-led integrative care.

Doctors of osteopathic medicine (DOs) represent more than 11% of the US physician workforce, and 15% of medical students in the US are training to be DOs [8]. One of the defining characteristics of osteopathic medical practice since its inception is the integration of OMM in a variety of clinical settings to alleviate somatic dysfunction, by itself, or in addition to patient education, medication and surgery in the care of the patient. However, the increasing prevalence of DOs in the US physician workforce has not led to a substantial increase in the use of OMM [7]. In order to address this gap in patient access to OMM, it is helpful to understand the factors that affect a DO's decision to use OMM. Barriers such as time constraints, lack of proficiency, insufficient reimbursement, and unsupportive work environments likely contribute to this underuse [6,9,10].

Little is known about the internal decision-making processes that shape a physician's choice to use OMM. This limited understanding hinders efforts to develop targeted solutions that could enable DOs to integrate OMM more consistently into practice. Prior research on this topic is sparse, and the scant research performed so far has been survey-based. The published articles do not report on the methodology used to create the survey questions, and there are likely unexplored themes in prior studies [7,9]. This indicates the need for additional primary research in this area.

The present study aims to explore how DOs make decisions regarding the integration of OMM into their clinical practice. Using in-depth interviews and thematic analysis [11], we examined how DOs understand and navigate the factors that shape their use of OMM. To our knowledge, it is the first to explore this topic through the lens of physicians' lived experiences. By generating new insights into the practical, professional, and systemic influences on OMM use, with this study we aim to inform efforts to reduce barriers and promote more consistent integration of OMM in patient care, ultimately advancing the role of non-pharmacological therapies in modern medicine.

## Methods

This project was approved by the Mayo Clinic Institutional Review Board on June 16, 2023 (case ID, 23–004379). Recruitment of interviewees began on the same day. Funding was provided through a Mayo Clinic Health System Research Internal Award grant. Funding was used for research support staff, transcription services, and primary researcher protected time. Because interviews were conducted remotely via Zoom, interviewees provided verbal informed consent prior to participation (S1 File). The use of verbal consent, including consent to use anonymized quotations in publications, was approved by the IRB (S2 File). Interviewees were not compensated for their participation.

Participants were recruited using purposive sampling through an email invitation sent to all DOs employed at Mayo Clinic and Mayo Clinic Health System (n = 184) asking whether they would volunteer to undergo a detailed interview about their attitudes and experiences with OMM. A research assistant contacted the volunteers and scheduled them for an interview with 1 of 3 investigators. Three investigators conducted the interviews: a male osteopathic family medicine faculty member with qualitative research experience (S.K.S.) and two female osteopathic family medicine residents (K.M.P., S.A.A.). All interviewers were trained in qualitative interviewing methods prior to the study.

Participants were informed of the study's purposes and that the research was being led by a team committed to advancing OMM scholarship. The interviewers were practicing osteopathic physicians and were perceived as peers by participants, which may have influenced openness during interviews. Interviewer assumptions and potential biases were discussed among the team throughout the research process to enhance reflexivity. Each of the 3 interviewers conducted an initial pilot interview. The results of the pilot interview were discussed with all authors, who agreed that the information gathered from the pilot interviews was adequate, and no changes were made to the interview script. Pilot interviews were not included in the final analysis.

The first interview was conducted on July 11, 2023. Interviews were conducted over Zoom (Zoom Video Communications, Inc.) using a qualitative semistructured interview guide [12]. Only the interviewer and participant were present during interviews, and no repeat interviews were conducted. Field notes were taken by interviewers during and after interviews to capture contextual impressions. Interview durations ranged from 8–40 minutes. Data saturation was discussed during analysis and deemed to have been reached when no new themes emerged in the final interviews. Transcripts were not returned to participants for review or comment. The final interview was conducted on August 29, 2023.

All interviews were audio recorded and transcribed verbatim. Personal information was redacted, and 3 investigators (S.K.S., K.M.P., and S.A.A.) analyzed each transcribed interview. A separate investigator (A.Q.) acted as a critical friend to enhance the quality and rigor of the analysis [13]. We recorded the age, sex, years in practice, degrees, specialty, and level of OMM use of each interviewee. Level of OMM use was rated on the following scale: high (most patients seen, dedicated referral system); medium (frequent, daily or almost daily, may or may not take referrals); low (occasionally, weekly or less, does not take referrals); none (does not perform OMM clinically). To analyze qualitative data, we employed abductive thematic analysis characterized by theoretical deduction and inductive features [14], which recursively influenced the construction of themes. This approach was grounded in a constructionist paradigm, characterized by ontological critical realism (i.e., reality as mind-independent) and epistemological subjectivism (i.e., meaning is created in the interaction between interviewer and interviewee). This methodology allowed us to explore the nuanced ways in which interviewee perspectives both shape and are shaped by their experiences with using OMM [15]. Results are presented in the form of aggregated themes emphasized by the embedment of anonymized quotes directly extracted from the transcripts. Results were reported according to the consolidated criteria for reporting qualitative studies (COREQ) checklist.

## Results

We received 76 responses from our survey (response rate 41.3%). Of those, 27 agreed to be contacted for further questions, and 16 agreed to complete an interview (Fig 1), providing a suitable sample to complete a thematic analysis [11,16]. No participants withdrew after agreeing to be interviewed. Respondents were between the ages of 33–66 (mean 42.3),

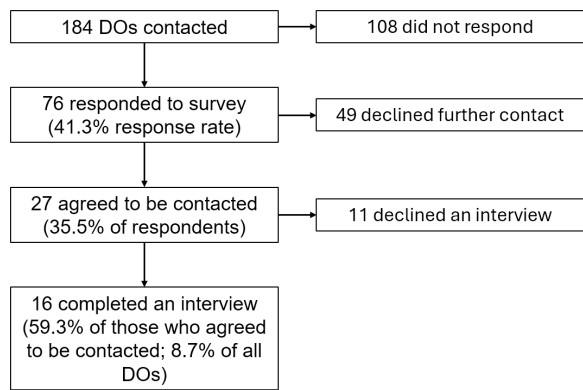

**Fig 1. Participant Flow Diagram. DO indicates doctor of osteopathic medicine.**

ranging from 3–32 years in practice (mean 12.0). The most represented specialty was family medicine, and most respondents reported either low (n = 7, 43.8%) or medium (n = 3, 18.8%) levels of OMM use, and approximately one third (n = 6, 37.5%) reported no OMM use clinically (Table 1).

Each interview transcript was independently coded by the two investigators who had not conducted the interview. Coding was done manually without the use of qualitative analysis software. Codes were generated from the interview data and iteratively organized into broader conceptual categories through team discussion. These categories were then synthesized into overarching themes using an abductive thematic analysis approach following a hierarchical process from codes to categories to themes. Transcripts and findings were not returned to participants for feedback.

All interviewees generally had favorable views of OMM regardless of their level of OMM use. They felt that OMM should be used more frequently than it is currently. This was expressed succinctly by the interviewee who said, "Too much demand, not enough supply" (P13). Interviewees observed that very few DOs practice OMM. They provided insight into what leads DOs to decide whether to offer OMM in their practice. We have organized these results around 6 key themes (Table 2):

1. The system they work in, including policies and leadership, shapes how much DOs feel empowered to incorporate OMM into their practice.

2. DOs differ in their feelings of capability when providing OMM.

3. OMM offers varying levels of utility depending on the attitudes of the DO and their practice environment.

4. The level of training experienced by the DO, especially during residency, affects their level of comfort and proficiency when providing OMM.

5. DOs recognize that the attitudes and beliefs of non-DOs play a role in their decision to provide OMM.

6. Providing OMM requires time and physical resources.

**System**

DOs provide OMM in the context of a health care system with policies, leadership, and environmental factors that may be more or less conducive to OMM. When interviewees discussed systems issues, they primarily focused on referrals, appointments, and support from peers and leadership. Regarding referrals, interviewees reported that good relationships with DO colleagues help referring providers trust that their patients will benefit from OMM. Several noted that referring

**Table 1. Demographics of interviewees.**

| Participant Number | Age | Sex | Years in practice | Degrees | Specialty | OMM use[a] |
|---|---|---|---|---|---|---|
| P01 | 33 | M | 11 | DO | Psychiatry | None |
| P02 | 34 | M | 8 | DO | Family medicine | Medium |
| P03 | 37 | F | 3 | DO | Neuropsychiatry | None |
| P04 | 37 | M | 4 | DO, MS | Family medicine | Low |
| P05 | 39 | F | 7 | DO | General surgery | None |
| P06 | 39 | M | 7 | DO, MS | Pediatric rheumatology | None |
| P07 | 40 | F | 9 | DO, MPA, MBBS | Family medicine | Medium |
| P08 | 40 | M | 10 | DO, MPH | Internal medicine | Low |
| P09 | 41 | F | 14 | DO | Family medicine | None |
| P10 | 42 | M | 8 | DO | Internal medicine/NMM | Medium |
| P11 | 42 | F | 16 | DO, MHA | Palliative care | Low |
| P12 | 43 | F | 11 | DO | Family medicine | Low |
| P13 | 44 | F | 14 | DO | Family medicine | Low |
| P14 | 47 | F | 16 | DO | Pediatric neurology | None |
| P15 | 52 | M | 22 | DO | Family medicine | Low |
| P16 | 66 | M | 32 | DO, OD, MS | Ophthalmology | Low |

Abbreviations: DO, Doctor of Osteopathic Medicine; MBBS, Bachelor of Medicine, Bachelor of Surgery; MHA, Master of Health Administration; MPA, Master of Public Administration; MPH, Master of Public Health; MS, Master of Science; NNM, neuromusculoskeletal medicine; OD, Doctor of Optometry; OMM, osteopathic manipulative medicine.

[a]OMM use: *High* (most patients seen, dedicated referral system); *Medium* (frequent, daily or almost daily, may or may not take referrals); *Low* (occasionally, weekly or less, does not take referrals); *None* (does not perform OMM in a clinical setting)

providers prefer easy referral mechanisms, standardized order sets, streamlined scheduling, and education on appropriate referrals in order to provide "ease of availability, ease of scheduling" (P03). Clinicians who refer for OMM want "options to refer locally" (P08), and they appreciate having partners in the same practice for internal referral. The limited availability of OMM appointments causes long wait times, "even getting people when you refer into the OMM clinic, it can be a month later" (P09). One family physician who provides OMM stated she doesn't use OMM as often as she would like because "it takes six months to get into see me" (P13). Many interviewees felt this was much too long, and that the ideal follow-up duration would be only a few weeks.

Several interviewees expressed a desire for autonomy in the amount and duration of their OMM visits; this was seen as a weak point within their practice environment. A family physician observed, "We need to allow for that flexibility of scheduling, and I think that is the biggest barrier" (P13). They reported it is easier to provide OMM during visits scheduled for this purpose rather than incidentally during other visit types, in part because "our schedules are packed" (P12). They use various models for scheduling these visits, such as dedicated half days or 1 appointment per day. If they allowed OMM to be scheduled during any visit, it could quickly overwhelm the schedule. One DO observed they "get so many referrals for osteopathic manipulation that we have to put session limits on them" (P13). The atypical schedules for OMM visits can present problems with centralized schedulers with limited knowledge of OMM referral services, leading to double bookings, visits at the wrong time, or visits that are too short. One family medicine physician lamented, "The time that it would take to really appropriately treat patients and trying to arrange [OMM visits] in a schedule that would fit within our clinic structure" would "be a big hindrance" (P02).

Interviewees who provide OMM valued support from colleagues to discuss cases, validate treatment course, or see patients for OMM when they are out of the office. They also appreciate having a strong regional advocate to represent their needs in leadership discussions. Many interviewees would like to see peer networking

**Table 2. Key factors underlying the decision to provide OMM.**

| System (policy and support) | Utility (interviewee's perception of the usefulness of OMM in general/within their practice) | Attitudes and Beliefs (interviewee's perception of others' attitudes about OMM/DOs) |
|---|---|---|
| 1. Referrals are a key part of helping patients find OMM services<br>◦ There is too much demand for the supply of OMM visits<br>◦ Referring providers have a hard time finding physicians who provide OMM<br>◦ Providers need confidence that the referral network will continue in order to make it a practice habit<br>◦ It is important to have good relationships between referring providers and DOs who provide OMM<br>◦ Referring providers need to trust that their patients will benefit from OMM<br>◦ They may need education on appropriate reasons to refer<br>◦ Providers desire to refer if mechanisms are in place<br>◦ Standardized order sets help providers refer intuitively<br>◦ OMM needs to be provided near enough for patients to attend<br>◦ It takes too long to get in to see physicians for OMM<br>2. Physicians who provide OMM need scheduling support<br>◦ Streamlined scheduling helps patients make appointments<br>◦ Scheduled visits may be easier than impromptu treatments provided during routine visits<br>◦ The number of OMM visits needs to be balanced against the number of regular clinic visits, especially for residents<br>◦ Physicians want autonomy in the amount and duration of their OMM visits; this was seen as a weak point within their practice environment<br>◦ Centralized schedulers have a limited knowledge of OMM<br>◦ Improper scheduling/ordering OMM can lead to problems such as double bookings or visits that are too short<br>3. Physicians who provide manipulation benefit from peer support<br>◦ They especially value having a strong regional advocate<br>◦ They value having colleagues to discuss cases with<br>◦ Others who provide OMM can validate the treatment course<br>◦ It helps to have partners to see patients for manipulation when they are out<br>◦ Physicians would like peer networking opportunities or support groups<br>4. Clinic staff help support OMM visits<br>◦ Rooming staff needs to understand what is required for OMM<br>◦ Physicians need to have enough rooming staff<br>◦ Scribes are helpful in improving physician efficiency<br>5. Leadership support plays an important role in facilitating OMM<br>◦ Leaders support physician autonomy and obtaining resources such as tables and rooms<br>◦ Poor understanding of OMM limits leadership support<br>◦ Leaders reduce barriers such as short visits<br>◦ They should include OMM in talent management and hiring decisions<br>◦ Physicians feel over-empaneled, and leaders make important decisions regarding empanelment<br>◦ Physicians feel unable to cap panels to reflect work spent doing primary care vs. OMM<br>6. Use of OMM needs to fit with other financial priorities<br>◦ Confusion with reimbursement and billing may lead some to use less OMM<br>◦ Billing and coding are easy in the electronic health record<br>◦ Billing support staff need appropriate knowledge of OMM billing and coding to make sure they are not downcoding<br>◦ Other procedures may reimburse better, limiting uptake by procedural specialties<br>◦ Physicians recognize that OMM needs to fit with the organization's bottom line | 1. Physicians who provide OMM see it as a valuable resource<br>◦ OMM helps avoid use of medications, especially opioids<br>◦ OMM is useful in acute and chronic illnesses, especially for musculoskeletal pain conditions<br>◦ The touch involved in performing OMM facilitates connection with patients<br>◦ OMM is a useful adjunct to other therapies<br>◦ OMM can be used in all life stages, including pregnancy, adults, children<br>◦ Physicians have positive personal experiences with OMM<br>◦ Many physicians use OMM on family members<br>◦ DOs find value in OMM even if they don't practice it themselves<br>2. The utility of OMM differs by specialty<br>◦ The patient context is important when deciding whether to use OMM<br>◦ OMM is useful in primary care<br>◦ If touch is not part of the specialty (e.g., psychiatry), OMM is not considered by the physician<br>◦ Procedural specialties may view it as less useful due to poor comparative revenue generation<br>3. The poor evidence base of OMM is a drawback when discussing OMM and considering whether to use it<br>◦ The evidence for OMM is mostly based on personal experience<br>◦ A poor research base generates skepticism among physicians, which makes it seem less important | 1. Recognition of OMM varies, affecting both patient demand and provider perception<br>◦ Even in healthcare people may not know what a DO is<br>◦ Healthcare colleagues may need education about what OMM is<br>◦ OMM is not well-recognized by patients in their region<br>◦ Patients and providers frequently compare OMM with chiropractic, or even confuse OMM with chiropractic<br>◦ Patients and providers have a more favorable view of OMM when they have experienced it themselves<br>◦ When patients know OMM is available, demand increases<br>2. Patients generally have favorable attitudes towards OMM<br>◦ OMM may be seen as a last resort<br>◦ Patients may seek out DOs for holistic care<br>◦ Patients may expect resolution of pain even when that isn't what's being offered<br>◦ Patients commonly express excitement<br>◦ Their experience of OMM is better if they find success<br>◦ A brief treatment can be a "sample"<br>◦ Patients appreciate that OMM helps them avoid medication side effects<br>◦ Some patients may view with stigma or skepticism<br>◦ They may not be comfortable with the idea of manual therapy<br>3. Providers' attitudes towards OMM tend to be positive, particularly for chronic pain referrals<br>◦ Many who know about OMM seek it out for their patients<br>◦ Provider attitudes may improve with education or positive experiences with OMM<br>◦ Referring providers often refer for chronic pain indications, potentially viewing OMM as a pain consult service<br>◦ Non-medication options for pain control can help address issues with opiate use<br>◦ Providers may view OMM with skepticism, seeing it as fringe medicine due to its poor evidence base |

*(Continued)*

| System (policy and support) | Utility (interviewee's perception of the usefulness of OMM in general/ within their practice) | Attitudes and Beliefs (interviewee's perception of others' attitudes about OMM/DOs) |
|---|---|---|
| **Capability** (physician ability/comfort) | **Training** (importance of training to continued use of OMM) | **Resources** (time and physical) |
| 1. Regular use prevents skill degradation<br>◦ Those who currently perform OMM need to see enough patients to maintain their skills (i.e., "use it or lose it")<br>◦ Endurance may be required as OMM can be physically strenuous<br>◦ Physicians who practice OMM need resources for continued medical education such as time and funds for travel, conference fees, etc.<br>2. Many DOs prefer using quick techniques<br>◦ Fascial Distortion Model (FDM), High-Velocity, Low-Amplitude (HVLA) and Muscle Energy (ME) techniques were viewed as quick and effective<br>◦ Many DOs don't feel comfortable with HVLA, which was seen as a more involved, technical skill with greater risk and difficulty<br>◦ Many DOs value techniques that can be performed during a physical exam<br>◦ OMM was contrasted with physical therapy, which can take more time<br>3. Enjoyment is a key factor in OMM use<br>◦ Physicians that use OMM enjoy doing it<br>◦ Physician that don't use OMM spoke fondly of times when they used to use it (e.g., during medical school)<br>◦ It gives them a feeling of unique service<br>◦ Some DOs just don't want to use OMM | 1. Most DOs report having adequate OMM training in medical school<br>◦ Many go into DO school for reasons other than OMM, including proximity, a DO mentor, or the osteopathic philosophy<br>◦ It is easier to get into osteopathic medical schools<br>◦ Medical school training gave the impression that OMM was separate from mainstream medicine<br>◦ Medical school OMM training was broad-based, training students to treat patient of all ages using many different techniques<br>2. Inadequate OMM training in residency was a major barrier to continued use<br>◦ People who didn't practice OMM in residency are unlikely to do so after residency<br>◦ Residents in programs without OMM training need to be proactive to find opportunities to treat<br>◦ The closer people are to training, the more likely they may be to practice OMM<br>◦ Residents using OMM helps them become comfortable with using it in practice<br>◦ They can be overwhelmed by OMM visits if they are not balanced appropriately with other responsibilities<br>◦ OMM mentors help motivate and teach others to use OMM in patient care<br>3. Practicing physicians can re-learn OMM skills<br>◦ Many DOs are out of practice but would be willing to perform more if they felt comfortable<br>◦ Mentors could help them re-learn skills<br>◦ They can attend courses to help re-train them in their skills<br>◦ Peers who provide OMM can help support a manipulation practice | 1. Physicians find OMM time-consuming amid busy schedules and diverse patient needs<br>◦ There are multiple extra steps involved in OMM<br>◦ Complicated pain takes more time, and this is often the reason for referrals<br>◦ Physicians feel like they already take care of too many problems<br>◦ Physicians have other tasks such as needing to focus on quality measures<br>◦ Patients expect physicians to fit multiple complaints into a single visit<br>◦ Many physicians are already running behind<br>◦ OMM visits need to be long enough<br>◦ Some physicians have to squeeze manipulation patients in between already scheduled visits<br>2. Physicians require sufficient physical resources<br>◦ A proper table is important for patient and physician comfort, especially one with adjustable height<br>◦ Performance of OMM requires a room that is large enough<br>◦ A clinic requires enough rooms for the number of physicians performing OMM |

Abbreviations: DO, doctor of osteopathic medicine; OMM, osteopathic manipulative medicine.

opportunities, such as an OMM interest group, to learn useful techniques, discuss patient cases, and advocate for each other. As one said, "It's helpful to actually have people around us that have a similar background, similar experiences, similar knowledge base… We offer something different than MDs, and it would probably just help us make connections" (P04).

In discussing how leadership affects the decision to use OMM, interviewees wanted their leaders to support physician autonomy and create an environment where OMM can flourish. They reported that their leaders generally allow practice of OMM and were helpful in providing resources such as tables, rooms, and appointments for OMM. However, as 1 interviewee stated, "we are just so woefully over paneled" (P09), and many felt that "there isn't an ability to necessarily cap their panels" (P03) to reflect the time they spent performing OMM. Interviewees wanted their leaders to facilitate OMM education for health care staff and help spread news of OMM practices through "general education to the masses" (P15). They also hoped their leaders would focus more on recruiting and retaining DOs to practice OMM. One interviewee felt if their leadership would hire somebody "three days a week for family med and one day a week for OMM," then "we'd get those people here" (P13).

Interviewees were mixed about whether reimbursement limited the amount of time they felt they could reasonably spend performing OMM. One family physician noted "there's pretty good reimbursement" (P02). This was especially true for procedural specialists such as surgeons. Those who perform OMM reported that billing is easy in the electronic health record, but billing support staff need appropriate training on OMM billing to make sure that coders do not overturn their billing. One physician who provides OMM noted the need to be "diligent in your practice to check the billings happening, your coding's not getting downcoded" or else "sometimes it doesn't get billed correctly" (P12).

## Capability

Interviewees varied in their self-assessment of their capability of providing OMM. Technical prowess was part of their assessment, but emotional responses appeared to be equally important. Enjoyment emerged as a key factor in the decision to provide OMM. Interviewees who use OMM reported that they enjoy doing it, while those who do not use OMM spoke fondly of times when they used to use it. A psychiatrist who does not use OMM observed that it was nonetheless "a vital part" of their medical school rotations (P01). One interviewee said, "I enjoy doing it, and I think I'm delivering a service that nobody else is delivering here" (P16). However, some reported not wanting to use OMM.

Many interviewees who do not perform OMM cited skill degradation as a primary reason. One family physician recognized they were using OMM "less and less in my practice," and noted, "I felt my skills were dwindling somewhat" (P09). Training and regular practice were seen as ways to prevent skill loss, maintain stamina, and stay up to date on coding and billing; however, many reported having to rely on funding to attend continuing medical education events. Interviewees who did not feel comfortable with OMM reported they would need further training before resuming practice, and a few expressed interest in this training. "I would like to [use more OMM]," noted one interviewee. "I just haven't done it in so long I feel that I would need a refresher" (P14).

Quick techniques were seen as especially helpful for busy clinicians. Interviewees described several osteopathic techniques, including fascial distortion model, high-velocity low amplitude and muscle energy. Fascial distortion model (FDM) was highlighted as especially fast, simple, and effective. "I heavily use FDM and then muscle energy and myofascial," stated one family physician (P13). Rapid technique enabled more frequent treatment, as one physician observed, "Even despite going to 30-mintue visits over the last year and half, I've been able to do even more OMT with the direct techniques like FDM and stuff" (P02). High-velocity low amplitude (HVLA) and muscle energy techniques were also highlighted as quick, though not as easy to learn. Many interviewees "don't feel comfortable" (P02) with high-velocity low amplitude techniques, which were seen as more involved technical skills with greater risk and difficulty. Many interviewees said they value techniques that can be performed during a physical examination.

## Utility

Whether or not they provide OMM, many interviewees reported seeing it as a valuable treatment tool that helps avoid use of medications, especially opioids. One interviewee observed, "You will always have stories of people being on like 10,000 medications, getting all these pain medications, and then you will do OMM and then they're not taking medications" (P04). Interviewees found OMM useful in acute and chronic illnesses, especially for pain. They valued how touching patients through OMM facilitates connection. OMM was offered to patients in all life stages, including pregnant women, children, younger adults, and the elderly. Interviewees reported positive personal experiences with OMM. Many said they use OMM on family members even if they do not use it on patients, such as the general surgeon who mentioned that "I use it at home on my family all the time" (P05).

Interviewees stated clinical specialty was important when deciding whether to use OMM. Subspecialists and primary care DOs agreed that OMM is most useful in primary care practice. One family medicine provider stated, "Primary care is the work horse of who does [OMM] (P09). One psychiatrist noted that, "Obviously, my biggest barrier is that it doesn't fit into my practice setting and specialty" (P01). While some interviewees who practiced OMM were not in primary care, several agreed that, "If anybody is specializing in anything other than family medicine, or internal medicine, or primary care, then they're not likely to use it" (P15).

## Training

For a few interviewees, the holistic nature of osteopathic medical schools played a central role their decision to choose to become a DO. As one noted, "I only applied to one osteopathic school—one medical school. It was an osteopathic medical school, and it was because of that contextual comprehensive holistic approach to care" (P13). A pediatric neurologist observed, "I liked the idea that it was a little bit more holistic, looking at more preventative factors and the person as a whole" (P14). Many others were attracted by the perceived ease of admission among DO schools relative to MD schools. "Only a handful of DOs truly want to participate in OMT," observed one participant. "For those people, it's probably their driving decision to go into the osteopathic field. I think, for some people, it's just about getting into any medical school they can, and then having to be DO or MD as long as they can become a physician" (P10). This concept was supported by a physician who stated, "The allopathic school put me on a wait list, so then I found an alternative" (P15). Several interviewees described choosing their medical school based off of location, as stated by a participant who stated he picked a DO school because, "Geographically, that's where I wanted to be" (P06).

Presence of a DO mentor and appeal of the osteopathic philosophy also influenced their choice, such as the general surgeon who stated, "My dad's partner was a DO, which I didn't realize he was any different than the MD… I really liked his partner as a person and the way he practiced medicine" (P05). However, some found that their medical school training gave the impression that OMM was separate from mainstream medicine. One family physician noted that, "There was sort of this approach to medical education that was like, here's medicine, and here's how to do your physical exam, and here's how to think about disease processes. Then here's OMT. The entire process did not really integrate OMT into medicine, and so I had this constant cognitive dissonance between [medicine and OMT]… There were times where it felt like it was pulling me from medicine, and it being part of medicine" (P13).

Several DOs reported receiving adequate OMM training during medical school, which prepared them to treat patients of all ages using various techniques. "I think as a medical student I had great exposure," remarked one interviewee. "As a fourth-year medical student, I had excellent exposure. I mean, I did it all the time" (P14).

In contrast, the primary training barrier reported by interviewees was inadequate OMM training during residency. As one remarked, "I learned OMM during med school, but I really didn't do a lot during residency because I didn't have supervision. It was pretty limited, and it remains really pretty limited" (P08). Another noted that when trainees don't practice OMM during residency, "They don't incorporate it or don't know how to incorporate it" (P07). Alternatively, one interviewee observed that, "If you saw it in residency and you were like, 'Hey, this really helped in this patient,' you would use it" (P05).

## Attitudes and beliefs

Interviewees reported OMM is not well-recognized by patients in their region, affecting both patient demand and provider perception. Even in their health system, people may not know what a DO is, including non-osteopathic physicians (especially specialists), schedulers, nurses, and leadership. Interviewees reported frequently observing patients and providers comparing OMM with chiropractic. Some interviewees were bothered by this comparison, while others welcomed it. One interviewee noted that when the comparison is brought up, "We get to have a conversation about how OMM is different than chiropractic and what are advantages or disadvantages for that" (P03).

Many interviewees observed that physician attitudes towards OMM tend to be positive, particularly for chronic pain indications. They often express excitement and are eager to refer. Attitudes differ by specialty and tend to improve with education and positive experiences with OMM. Staff wellness events were seen as a way to provide positive experiences and education, as one interviewee mentioned, "If [staff] experience it personally, that might probably help them spread the word" (P04).

Likewise, interviewees who perform OMM reported patients have a more favorable view of OMM when they have experienced it themselves. They stated that some patients seek them out for holistic care. One interviewee observed, "A lot of people come to me knowing what a DO is, because they've heard of them or they've had experience before, so when I offer [OMM], they're excited" (P07). Another mentioned, "When you can take the time to really explain what it is you're doing and why you're doing it, there's very few that actually then don't want to proceed" (P11). Many reported that patients seem to have varying expectations of what OMM can do, such as expecting resolution of pain even when that is not offered.

Interviewees expressed that providers seemed more likely than patients to view OMM with skepticism. A pediatric rheumatologist mentioned, "I think the evidenced-based part is the question mark, right?" (P06) An internal medicine physician added, "There's some definite fringe things in OMM that probably need to go away. I remember learning about craniosacral therapy when I was in med school. We're not moving the bones. There's no good biologic basis for that, so I think we need to get away from some of that stuff that isn't helping us with our MD colleagues, getting acceptance. Some of that's in our court as a profession" (P08). This interviewee continued, "The research was crap. Really, it was. It was garbage" (P08).

## Resources

Interviewees reported that time and physical resources strongly affected their decision to provide OMM. Performing OMM was seen by interviewees as time-consuming amid busy schedules and diverse patient needs. The addition of OMM was seen as taking time that physicians may not have. "It just always takes more time than any other routine exam that I would do" (P16). "I was contacted before if I wanted to increase my manipulation, I said yes. Did it increase? Did I get more time? No" (P07). "The time that it would take to really appropriately treat patients and trying to arrange those in a schedule that would fit within our clinic structure, I think would be a big hindrance to trying to get something like that going more consistently" (P11).

Interviewees highlighted multiple extra steps involved in OMM, including:

1. Explaining the procedure, getting buy-in, and obtaining consent (this may take more time than the procedure itself)

2. Obtaining additional history

3. Performing the procedure

4. Documenting and coding the procedure

Interviewees felt that they already take care of too many problems; as one said, "I think we're all overbooked and overworked and it's hard to see every patient and it's hard to get to your meetings and then also get home" (P05). Visits for

OMM need to be long enough. Many interviewees reported that sessions should be 20–40 minutes depending on the context. Thirty minutes appeared to be adequate for most interviewees. Some reported having to find time for OMM between already-scheduled visits.

Interviewees who perform OMM reported needing access to an OMM table and a large enough room in which to perform OMM. They felt it was important for the table to have adjustable height and preferred tables with a face opening. An adjustable-height table was seen as important for patient and physician comfort. One physician stated that, "It hurts my back to do OMT on [fixed-height] tables" (P04). Also, a clinic requires enough rooms for the number of physicians performing OMM. "The biggest hinderance on that," noted one DO, "has been we only have one procedure room between myself and [my partner]" (P02).

## Discussion

This study explored the perspectives of DOs practicing within a large, multi-specialty health system to better understand the factors that limit or enable OMM use. OMM is recognized for its holistic approach to patient care and aligns with the increasing demand for nonpharmacologic treatments, particularly in the management of musculoskeletal disorders [17,18]. It remains underutilized, however. Our data suggests that this underuse is not primarily driven by skepticism of osteopathic physicians, but rather by a combination of structural, educational, and cultural barriers.

Our study's central findings underscore a complex interplay of system-level factors influencing the integration of OMM into routine clinical practice. This includes scheduling constraints, clinic infrastructure, leadership priorities, and the availability of time within standard consultation sessions. These resource limitations are compounded by systemic issues, such as inflexible scheduling systems and inadequate referral processes, which hinder the seamless integration of OMM services. In this environment, leadership support is not merely a facilitator, but a precondition for systematic practice of OMM. Leaders determine resources such as appointment length, rooming staff, table access, and panel size. This type of hierarchy is typical of large organizations and may limit opportunities for OMM unless explicit institutional support is in place. This likely contributes to the perception that DOs must be "allowed" by the institution to use OMM.

Our results point to several actionable strategies to enhance the integration of OMM into large clinical systems. Structural adaptations include allowing physicians to schedule dedicated time for OMM, ensuring access to appropriate physical space and tables, implementing standardized order sets to streamline referrals, and developing systems of patient and provider education about the benefits of OMM [19]. Enhancing awareness could drive demand and acceptance, thereby encouraging health care systems to better accommodate OMM in their service offerings. Addressing these challenges through targeted interventions, such as streamlining scheduling processes, expanding training opportunities, and improving internal referral systems, could improve patient access to OMM, thereby enhancing patient outcomes [10].

Furthermore, our results reveal that while there is a general recognition among DOs of the utility of OMM, varying levels of capability and differing attitudes towards the practice significantly affect its use. Interviewees acknowledged the effectiveness of OMM, yet many described a steep decline in skill and confidence following graduation from medical school, particularly when residency training lacked dedicated OMM instruction. To address this, residency programs should consider implementing formal OMM curricula taught by faculty with the skills to teach and practice OMM. By pursuing or maintaining ACGME Osteopathic Recognition, programs may be especially well-positioned to incorporate hands-on supervision, protected time for OMM practice, and mentorship from qualified faculty.

In addition to improving residency training, there is a need for ongoing professional development to support DOs who wish to maintain or return to OMM practice [20]. Interviewees expressed a desire for refresher training, opportunities to shadow skilled colleagues, and access to continuing medical education courses focused on manual techniques. Supporting these efforts through protected time, funding, and peer mentorship could help re-engage DOs who wish to provide OMM but currently lack confidence or proficiency.

These results, while consistent across interviewees, represent only a select portion of DOs at a single institution where patients typically see DOs for conventional care rather than OMM per se. The identity of our interviewees as physicians appears to be more salient than their identity as manipulative medicine providers. Interviewees were primarily engaged in general clinical care, and those who offered OMM did so as part of routine patient care rather than as a referral-based specialty service. As such, these findings may be most applicable to DOs practicing in large, multispecialty settings. Of course the opinions of DOs within our institution who were not interviewed and DOs at other institutions likely differ in ways that cannot be addressed by this study. However, prior research in other settings has provided similar findings [9]. For example, DOs specializing in emergency medicine report a similar set of difficulties, including time constraints, unproven benefits, reimbursement issues, physician insecurity about OMM skills, and physician disinterest [10].

Future research should build on the foundational insights generated by this qualitative study, including extending this work to include a wider range of practice environments. The themes identified in these interviews also offer foundational support for the development of other quantitative or qualitative instruments. For example, future survey-based research could use these findings to inform item generation and domain structure to enhance the content validity of tools assessing attitudes, barriers, and facilitators related to OMM implementation. In this way, these results lay the groundwork for future mixed-methods or longitudinal investigations that could further clarify how OMM becomes integrated into clinical care.

While discussing the barriers facing implementation of OMM, interviewees provided insight into potential solutions. Many of these solutions center around finding time for OMM and obtaining support from leadership. Leadership support, in turn, facilitates resolution of many of the barriers reported by the interviewees [21]. This research enables individuals such as health care leaders to target the barriers most impactful to DOs providing OMM.

## Conclusion

When interviewed about the challenges of integrating OMM into their practice, DOs report limitations including time constraints, scheduling difficulties, limited referral options, and patient awareness. They recognize the value of OMM but often struggle to incorporate it into routine care, particularly in specialties where it may not be a primary treatment modality. They report that systemic barriers, such as patient panel management and practice autonomy, hinder OMM implementation. Stakeholders can address these challenges by streamlining scheduling processes, enhancing OMM training, strengthening referral processes, increasing patient education and awareness, advocating for practice integration, and empowering physician autonomy. Overcoming barriers to OMM use and promoting its integration into mainstream health care practice can improve patient access to this valuable treatment modality.

## Supporting information

**S1 File. Oral consent script.**
(DOCX)

**S2 File. IRB exempt form.**
(PDF)

## Author contributions

**Conceptualization:** Stephen K Stacey, Kelsie M. Prins, Sydney A. Anderson, Alessandro Quartiroli.

**Data curation:** Stephen K Stacey, Kelsie M. Prins, Sydney A. Anderson.

**Formal analysis:** Stephen K Stacey, Kelsie M. Prins, Sydney A. Anderson.

**Funding acquisition:** Stephen K Stacey.

**Investigation:** Stephen K Stacey, Kelsie M. Prins, Sydney A. Anderson.

**Methodology:** Stephen K Stacey, Sydney A. Anderson, Alessandro Quartiroli.

**Project administration:** Stephen K Stacey.

**Resources:** Stephen K Stacey.

**Supervision:** Stephen K Stacey, Alessandro Quartiroli.

**Validation:** Stephen K Stacey, Alessandro Quartiroli.

**Visualization:** Stephen K Stacey, Alessandro Quartiroli.

**Writing – original draft:** Stephen K Stacey, Kelsie M. Prins, Sydney A. Anderson.

**Writing – review & editing:** Stephen K Stacey, Kelsie M. Prins, Sydney A. Anderson, Alessandro Quartiroli.

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
