## [Decision Letter · Decision Letter 0]

17 Jun 2025

Dear Dr. Stacey,

We look forward to receiving your revised manuscript.

Kind regards,

Jenny Wilkinson, PhD

Academic Editor

PLOS ONE

Journal Requirements:

2. In this instance it seems there may be acceptable restrictions in place that prevent the public sharing of your minimal data. However, in line with our goal of ensuring long-term data availability to all interested researchers, PLOS’ Data Policy states that authors cannot be the sole named individuals responsible for ensuring data access (http://journals.plos.org/plosone/s/data-availability#loc-acceptable-data-sharing-methods).

Reviewers' comments:

Reviewer's Responses to Questions

**Comments to the Author**

1. Is the manuscript technically sound, and do the data support the conclusions?

Reviewer #1: No

Reviewer #2: Partly

2. Has the statistical analysis been performed appropriately and rigorously?

Reviewer #1: N/A

Reviewer #2: N/A

3. Have the authors made all data underlying the findings in their manuscript fully available?

Reviewer #1: No

Reviewer #2: No

4. Is the manuscript presented in an intelligible fashion and written in standard English?

Reviewer #1: Yes

Reviewer #2: Yes

Reviewer #1: 1. Patient awareness of OMT was not assessed in the interviews or surveys of the doctors and should not therefore be part of the conclusion which should be derived from the data gathered in this study.

3. The transcripts themselves constitute the data and are not readily available to readers or researchers.

The first paragraph is very similar to another manuscript submitted for publication from the Mayo clinic, using surveys of MDs and DOs. Authors should vary the introductory paragraphs to distinguish the two manuscripts.

Also, there are several references about a gap, of different types and breadths, which is an overused term in this manuscript.

More suggestions are in the attached file.

Reviewer #2: The subject is particularly interesting, and the choice of interviews seems consistent with the topic.

However, the paper needs some major improvement to be considered for publication.

Firstly, I am surprised that the study design was primarily a mixed method (see supporting information, IRB exempt : Full Study Title: “A Mixed Methods Analysis of Barriers to Osteopathic Manipulative Medicine”), but the authors only published the qualitative part. ICMJE recommends publishing all data and analysis of the same study in one paper, not in several ones.

The paper should be written according to the COREQ checklists (see Equator). This is not the case here, and the information is poorly organized and hard to follow. In addition, a lot of information is missing that is mandatory according to COREQ and to be able to reproduce the study. The following information must be added.

In methods:

- What were the researcher’s credentials? E.g. PhD, MD

- Participant knowledge of the interviewer

- What was their occupation at the time of the study?

- Was the researcher male or female?

- What experience or training did the researcher have?

- What did the participants know about the researcher? e.g. personal goals, reasons for doing the research

- Was anyone else present besides the participants and researchers during the itw?

- Were repeat interviews carried out? If yes, how many?

- Were field notes made during and/or after the interview?

- The interview guide must be included.

- The coding tree must be included.

- Were transcripts returned to participants for comment and/or correction?

- What software, if applicable, was used to manage the data?

- Did participants provide feedback on the findings?

In results:

- Data saturation must be discussed

- The authors used the abductive thematic analysis so the Thematic Network Analysis figure must be included.

- What was the duration of the interviews?

- The theme covers (percents for each theme and each participant).

It is not clear if the pilot interviews have been added in the final analysis or not:

- If yes, authors must remove the pilot interviews from the analysis: pilot data cannot be added, they are done to test, not to analyse.

- If not, authors must detail the recruitment, the case data, etc. of these DO.

Authors said: “3 investigators (S.K.S., K.M.P., and S.A.A.) analyzed each transcribed interview. » Each investigators analyzed all 16 itw or only some?

There are not enough interviewees’ quotes. The authors do not provide verbatim transcripts, so there is a need for many more quotes to know whether these are really the respondents' ideas or those of the authors. At least, one or two complete quotes must be added after each paragraph. For example, for System theme, the first two paragraphs do not quote any itw.

What's more, the rare quotes are not accompanied by the interviewee's number. So we don't know the profile of the DO giving this or that information.

In introduction, authors say that solutions need to be put in place, and I would have liked to have found some in the discussion. The discussion needs to be completed. Actually, it simply repeats the results without really delving into literature. For example, the fact that the interviewees were mainly MD from the Mayo clinic introduces a selection bias. I assume that the patients who come to the clinic are not the same patients as those who come to family practice where the practice of OMM will be different and easier. The responses give the feeling that the interviewed DO do not take care of patients who have come for osteopathic treatment but who have come for medical treatment. This is confirmed by the fact that DOs need their Leadership's approval to practice OMM. This in fact affects the other themes, such as Capability (if they don't practice much, they lose confidence in themselves). All this needs to be discussed. The results are only applicable to osteopaths practicing in clinics as MD clinicians. One of the respondents pointed out that in family medicine or primary care, OMM would be much more widely used. The fact that samples were ODs in a clinic setting is barely mentioned, although this should be specified in the title. Moreover, the authors compare the study to an emergency department study (the study states this in its title).

Nor is there any discussion of the study's biases and limitations.

There are few other problems in the paper.

The authors talk about solutions in the introduction as one of the objectives, but no concrete solutions are provided and there are only a few very general sentences at the end of the discussion. The solutions need to be developed to make the work interesting.

Indeed, the objective is unclear. Several objectives are presented. Is to explore “the perspectives of DOs through indepth interviews and thematic analysis of their decision-making regarding the integration of OMM into their clinical practice” ou to provide “solutions to barriers they identified based on their training and professional experiences”? Both objectives are present in introduction. Since no solution are provided, the second sentence has to be modified.

In the background, it would have been good to temper the efficacy of osteopathy for chronic pain, depending on the type of pain and treatment. Some studies mention the absence of efficacy and the meta-analyses cited by the authors are not clear-cut on efficacy either or focus on specific patients. For example, Franke 2017 focuses on pregnant and post-partum women, which does not seem too relevant as a reference for chronic pain. Licciardone et al. is not a systematic review but a narrative review and cannot support efficacy.

The authors wrote: “However, the increasing prevalence of DOs in the US physician workforce has not led to a substantial increase in the use of OMM.” Since it is the reason why they ran their study, a reference must be added.

The authors wrote: “Barriers such as time constraints, lack of proficiency, insufficient reimbursement, and unsupportive work environments likely contribute to this underuse. » So they answered the question before doing the study?

Table 2. “Training and regular practice” is displayed in Capability theme and not in Training theme. Why? Ces 2 themes véhiculent les mêmes idées: “level of comfort and proficiency » and « their feelings of capability when providing OMM” looks like similar.

It's a bit the same for the System and Resources themes, where the ideas of organizing time, having rooms available in the clinic, having a table, etc., stand out in both.

**Do you want your identity to be public for this peer review?** For information about this choice, including consent withdrawal, please see our Privacy Policy

Reviewer #1: **Yes: ** Michael Seffinger, D.O.

Reviewer #2: No

---

## [Decision Letter · Decision Letter 1]

29 Jul 2025

Barriers to Osteopathic Manipulative Medicine Use: A Qualitative Study of Physician Attitudes and Experiences

PONE-D-25-16174R1

Dear Dr. Stacey,

We’re pleased to inform you that your manuscript has been judged scientifically suitable for publication and will be formally accepted for publication once it meets all outstanding technical requirements.

Kind regards,

Jenny Wilkinson, PhD

Academic Editor

PLOS ONE

Additional Editor Comments (optional):

Reviewers' comments:

Reviewer's Responses to Questions

**Comments to the Author**

Reviewer #1: All comments have been addressed

2. Is the manuscript technically sound, and do the data support the conclusions?

Reviewer #1: Yes

3. Has the statistical analysis been performed appropriately and rigorously?

Reviewer #1: N/A

4. Have the authors made all data underlying the findings in their manuscript fully available?

Reviewer #1: Yes

5. Is the manuscript presented in an intelligible fashion and written in standard English?

Reviewer #1: Yes

Reviewer #1: (No Response)

**Do you want your identity to be public for this peer review?** For information about this choice, including consent withdrawal, please see our Privacy Policy

Reviewer #1: **Yes: ** Michael Seffinger, D.O.

---

## [Editor Report · Acceptance letter]

PONE-D-25-16174R1

PLOS ONE

Dear Dr. Stacey,

I'm pleased to inform you that your manuscript has been deemed suitable for publication in PLOS ONE. Congratulations! Your manuscript is now being handed over to our production team.

Kind regards,

on behalf of

Dr Jenny Wilkinson

Academic Editor

PLOS ONE